# Effect of Contamination by Phosphate Mining Effluent on Biocrust Microbial Community Structure and Cyanobacterial Diversity in a Hot Dry Desert

**DOI:** 10.3390/microorganisms13112580

**Published:** 2025-11-12

**Authors:** Ali Nejidat, Damiana Diaz-Reck, Eli Zaady

**Affiliations:** 1Department of Environmental Hydrology and Microbiology, Zuckerberg Institute for Water Research, The Jacob Blaustein Institutes for Desert Research, Ben-Gurion University of the Negev, Sede Boqer Campus, Midreshet Ben-Gurion 84990, Israel; 2Katif Research & Development Center, The Ministry of Innovation, Science and Technology, P.O. Box 100, Netivot 8771002, Israel

**Keywords:** biological soil crusts, environmental disturbance, ephemeral streams, microbial diversity, mining effluent impact, Negev Desert

## Abstract

This study investigates the long-term effects of catastrophic phosphate mining effluent contamination on the biocrust microbial community structure in sections of the ephemeral Ashalim Stream, Negev Desert, Israel. Microbial communities were characterized using next-generation sequencing of 16S rRNA gene fragments, conducted 1.5 and 5 years after the contamination event, across five stream strips. Samples from the nearby, uncontaminated Gmalim Stream served as controls. Biocrusts from Ashalim showed higher relative abundances of the phyla Actinobacteria and Firmicutes compared to Gmalim, whereas Chloroflexi were more prevalent in the controls. At the genus level, *Blastococcus*, *Bacillus*, *Massilia*, and *Noviherbaspirillum* were more abundant in the Ashalim strips, while *Flavisolibacter*, *Segetibacter*, and *Rhodocytophaga* were more abundant in the controls. Notably, genera within the Cyanobacteria phylum accounted for only 0.0–2.0% of sequences in Ashalim samples versus 2.5–20% in controls. The filamentous *Leptolyngbya*, *Tychonema*, and *Trichocoleus* genera were the most dominant cyanobacteria in all samples. The Nitrogen-fixing cyanobacterial genera *Scytonema* and *Nostoc* were consistently detected in Gmalim, but only in trace numbers in certain Ashalim sites. The results from both sampling campaigns support the hypothesis that phosphate effluent contamination had a profound impact on biocrust microbial community structure and function. In particular, the marked reduction in Cyanobacteria suggests a long-lasting disruption that may substantially hinder the natural ecosystem rehabilitation.

## 1. Introduction

The ecosystems of arid and semi-arid regions, characterized by limited nutrient availability and moisture scarcity, are highly vulnerable to disturbances caused by anthropogenic activities and climate change [1,2,3]. These regions cover approximately 40% of the Earth’s terrestrial surface, and biological soil crusts [biocrusts] which occupy the spaces between vascular plant patches in deserts account for about 12% of the global terrestrial surface [4]. These are topsoil layers composed of microphytic communities consisting of cyanobacteria, green algae, lichens, mosses, and other microorganisms bound together and to soil particles by a mucilaginous exudate [5]. These communities reduce soil erosion, influence the soil’s hydrological properties [6,7,8,9,10], and, importantly, provide many of the ecosystem services that higher plants supply in more mesic environments, such as primary production and soil enrichment through C and N fixation [11,12,13]. Biocrusts are particularly vulnerable to climate change and land disturbances [14,15], and when damaged, may require 5–20 years to recover [16]. For instance, environmental modeling by Rodríguez-Caballero et al. [10] predicted a 25–40% decline in biocrust cover over the following 65 years as a result of anthropogenic activities and climate change.

Among anthropogenic activities, those associated with mining industries have particularly deleterious effects on biocrusts worldwide. These industries supply essential raw materials for human needs, including oil and gas for energy, fertilizers for agriculture, and various minerals for daily use [17]. Many resource-rich regions, however, are in desert areas, making them vulnerable to mining operations. Accidental spills from these industries release contaminants that persist in and severely damage the surrounding fragile desert ecosystems [18,19,20,21,22,23,24]. Here, we focus on a specific instance of such destruction; in June 2017, the wall of a reservoir containing wastewater from a phosphate mining plant failed, releasing approximately 150,000 cubic meters of industrial effluent into the ephemeral Ashalim Stream, located in the Ashalim Nature Reserve in the Negev Desert, Israel. This event contaminated both the stream’s mainstream and its margins [25]. The wastewater originated from nearby phosphate-processing facilities, where phosphate rocks are converted into phosphorus fertilizers, and contained elevated levels of phosphorus and sulfate compounds, heavy metal residues, fluorine, and various minerals [26,27]. Beyond introducing hazardous chemicals, the wastewater surge also inflicted severe physical damage on the biocrust.

Restoring the biocrust is essential for rehabilitating the stream ecosystem. Accordingly, this study aimed to examine the long-term effects (up to five years) of the 2017 contamination event on the microbial community structure of the biocrust in the Ashalim Stream, compared to the undisturbed biocrust in an adjacent, unaffected stream. Our initial hypothesis was that the massive influx of phosphate mining effluent would exert lasting impacts on the biocrust microbiome, particularly affecting the abundance and diversity of cyanobacteria, a key component of biocrust structure and function.

## 2. Materials and Methods

### 2.1. Site Description and Sample Collection

The Ashalim stream basin (66 km^2^) is situated between the Judean and Negev Deserts and serves as an important ecological corridor connecting these two regions. The stream can be divided into two main geomorphological units: a sandy upstream section and an alluvial downstream section. The sandy unit, which receives runoff from the Rotem Plain, is relatively wide and characterized by coarse-grained sandy soils. In contrast, the lower alluvial unit contains deeper soils typical of depositional environments. The basin is characterized by a hyperarid climate, with mean annual precipitation of approximately 70 mm in the upstream section and about 40 mm in the lower section, occurring mainly between November and April. Additional details regarding the study area are provided in Kerem et al. [25].

In June 2017, approximately 150,000 cubic meters of phosphate mining industrial wastewater contained in a reservoir situated in the upper sandy section of the Ashalim stream was accidentally released into the stream and flowed down to the Dead Sea. The chemical composition of the wastewater in the reservoir before the spill was documented by Tzoar [27]. Among other chemicals, it contained (in mg m^−3^): sodium, 1349; chlorine, 1180; calcium oxide, 11,707; sulfate, 33,223; silicon dioxide, 24,790; fluorine, 14,608; and phosphorus pentoxide, 219,445; in addition to a group of trace elements including lead, arsenic, nickel, molybdenum, copper, chromium and vanadium at lower concentrations [27].

Biocrust samples were collected from two sites—designated Ash1 and Ash2—in the sandy unit of the Ashalim stream: the first sampling was conducted in November 2018, 17 months after the spill (first sampling), and in July 2022, five years after the spill (second sampling). The Ash1 site (31°04′27.8″ N 35°13′37.1″ E) is situated in close vicinity of the damaged reservoir, and the Ash2 site (31°04′41.6″ N 35°13′56.4″ E) is downstream from Ash1. A sandy section of the Gmalim stream (31°05′32.6″ N 35°13′48.1″ E), about 1.5 km north of the Ash1 site, was chosen as a control, since this stream was not exposed to the contamination event and thus has undisturbed biocrusts typical of the area. In the Ashalim sampling area—due to the contamination—five different soil cover strips (henceforth strips) were identified along a cross-section from the bottom of the channel to its top. One-centimeter-deep topsoil surfaces (henceforth biocrust samples) were carefully collected from the five strips (Figure 1): stream bed (designated G); dark soil surface (D); bright soil surface (L); organic matter surface (O); and stream bank surface (N). In the Gmalim stream, biocrust samples were collected from three strips: stream bed (G); shrub area (S); and upper stream bank area (N). Four crust samples (composites of three subsamples) were collected from each strip at a distance of at least 5 m from one another.

### 2.2. Chemical Characterization of Biocrust Samples

Biocrust samples were sieved (<2.0 mm), and subsamples were used for the following analyses: pH and electrical conductivity (EC) were measured using a TDS/conductivity meter in a 1:2 (5 g soil per 10 mL) mixture of soil and distilled water. To determine inorganic nitrogen in the soil samples, 2.5 g soil were extracted in 10 mL of 1 M KCl (for 60 min) and filtered through a 0.45 µm filter. Total ammonia nitrogen (TAN = NH_4_^+^ + NH_3_) was determined by the Nessler method [28], and nitrate (NO_3_^−^) was determined by the second derivative method [29]. For ICP analysis, dry soil (0.5 g) was heated at 450 °C for 6 h to burn off the carbon. Thereafter, 10 mL of 1 N HNO_3_ were added, and the mixture was evaporated slowly to dryness on a hot plate. Subsequently, 10 mL of 1 N HCl were added to dissolve the residue, the solution was heated to near boiling, and 50 mL of water was added to the hot liquid. The liquid was filtered, and elemental concentrations were determined by ICP OES (Varian 720 ICP-OES, SPECTRO Analytical Instruments, Klave, Germany).

### 2.3. DNA Extraction, qPCR Assays and High-Throughput Sequencing

DNA was extracted from 0.5 g of soil using the PowerSoil^TM^ DNA Isolation Kit (MO BIO Lab. Inc., Solana Beach, CA, USA), according to the manufacturer’s instructions. DNA aliquots were stored at −20 °C. qPCR enumeration of the *nifH* gene, encoding the dinitrogenase reductase component of the nitrogenase enzyme that catalyzes N_2_ reduction to ammonium [30], was performed using CY81F (5′-GYGCTGTNGAAGATATWGA AC) and CY226R (5′-GCCGTTTTCTTCCAAGAAGT T) PCR-primers targeting the cyanobacteria *nifH* gene [31]. The CYA359F (5′-GGGGAATY(=C/T)TTCC GCAATGGG)and CYA781R (5′-GACTACW(=A/T)GGGGTATCTAATCCCTTT) PCR primers [32] were used to amplify cyanobacterial-specific 16S rRNA gene fragments from samples of the second sampling (in 2022). qPCR reaction mixtures contained: 12.5 µL of reaction mix (KAPA SYBR FAST qPCR Kit Master Mix) (Kapa Biosystems, Woburn, MA, USA), 0.5 µL of each primer (200 nM final concentration), 3 µL of template DNA, and 8.5 µL of doubly distilled water in a total volume of 25 µL. Melting curves (72–95 °C) showed only one peak for all qPCR reactions. Calibration curves were prepared by 10-fold serial dilutions (10^3^–10^9^ copies) of plasmids containing a corresponding gene fragment with *R*^2^ > 0.975 and PCR efficiencies of 90–111%. Amplifications were performed in a Rotor-Gene 6000 (Corbett Research, Sydney, Australia).

To determine the bacterial community structure, the DNA extracted from two biocrust samples were combined. Thus, for each stream strip two separate amplicons were generated. 16S rRNA gene fragments (V3–V4 regions) were PCR amplified using the primers 341F (5′-CCTACGGGAGGCAGCAG) and 806R (5′-GGACTACHVGGGTWTCTAAT) [33,34] attached to the linkers, 5′-ACACTGACGCATGGTTCT ACA and 5′-TACGGTAGCAGAGACTTGGTCT, respectively. Next-generation amplicon sequencing and microbiome bioinformatics were performed (as a service) by the Genomic and Microbiome Core Facility at Rush University, Chicago, IL, USA. Next-generation amplicon sequencing was performed on an Illumina MiSeq sequencer [35]. Microbiome bioinformatics was conducted with the QIIME2 2021.11 platform [36]. Raw sequence data were assessed for quality using FastQC and merged using PEAR. Merged sequences were quality filtered using the q2-demux plugin, followed by denoising with DADA2 [37] (via q2-dada2). Primer adapter sequences were removed using the cutadapt algorithm. Taxonomy was assigned to amplicon sequence variants (ASVs) by using the q2 feature classifier [38] and annotated to 25 phyla, showing various abundances in the different samples. The raw reads of the soil microbiomes were deposited under BioProject numbers https://www.ncbi.nlm.nih.gov/bioproject/PRJNA952203 (Raw sequence reads; Registration date: 4 April 2023) for the 2018 samples, and https://www.ncbi.nlm.nih.gov/bioproject/PRJNA960943 (Raw sequence reads; Registration date: 24 April 2023) for the 2022 samples in the NCBI sequence read archive (SRA).

## 3. Results

### 3.1. Biocrust Chemical Characteristics

Table 1 presents selected chemical properties of the biocrust samples. The pH values measured 17 months after the contamination event (first sampling) at the two Ashalim sites, particularly in the streambed strip, were lower than those of the control samples (except for one control streambed value). Although the pH values increased somewhat five years after the event, they remained lower in the streambed samples and in the strip immediately above the streambed compared to the upper strips and the control samples. The EC values of all the Ashalim samples were higher than those for the corresponding strips in the control. In general, the EC values were markedly lower for the five-year sampling compared to the 17-month sampling. It should be noted that EC values remained high after the contamination event, even though the stream was flushed with unknown amounts of fresh water immediately after the spillage. Furthermore, five years after the event, the EC values continued to be higher than the controls, although the area had experienced five major rainfall events and runoff during the 2019 rainy season and during one of those events (5 March 2019) the water level reached as high as 140 cm. In line with the composition of the contaminating wastewater, the Ashalim samples contained higher concentrations of phosphorus and sulfur than the control samples, as determined by ICP analysis (Appendix A). In general, the total amounts of all the minerals determined by ICP were higher in the Ashalim soil strips than in the corresponding control strips (Appendix A). The lower mineral contents of the samples from the lower strips vs. the upper strips, as reflected by the lower EC values (Table 1), were due to the flushing of the stream and the rain events.

Biocrusts are the main source of fixed nitrogen in hot deserts [4]. The results of the two sampling campaigns of the Ashalim biocrust strips and the control strips for inorganic nitrogen forms did not show any clear patterns.

### 3.2. Bacterial Diversity in the Biocrust Strips: Phylum Level

Figure 2 and Figure 3 depict the phyla detected in the biocrust strips based on their relative abundance during the first (Appendix A) and second (Appendix A) samplings. Actinobacteria, Proteobacteria and Bacteriodetes were the most dominant phyla in all samples, and, on average, they accounted for 60–72% of the sequences. In the first sampling, the Actinobacteria phylum was more abundant in most of the Ashalim samples than in the control samples, while in the second sampling the values were comparable. In Ashalim samples the Proteobacteria phylum was more abundant in the streambed strips (38–46%) than in the upper strips, and 33% in the control. The abundances of Bacteriodetes were similar in control and Ashalim samples, with higher abundances in the lower strips of the streams. In addition to these, other phyla had shown a differential abundance between Ashalim and control samples. On average, Firmicutes bacteria constituted 9.7 ± 5.3% and 5.6 ± 3.8% (not including an abundance of 49% in one strip of the first sampling) in the Ash1 and Ash 2 samples, respectively, and 2.5 ± 1.9% in the control samples. In contrast, the Chloroflexi constituted 10.4 ± 4.2% of the control samples and 4.1 ± 2.6% and 14.5 ± 3.9% of the lower three and upper two Ashalim strips, respectively. The Acidobacteria phylum was more abundant in the control samples (2–4.5%) than in all Ashalim samples (0–1%). The Cyanobacteria phylum, the key component of the biocrust community, was clearly more abundant in the control samples (2.5–20 than in Ashalim samples (0–2%).

Principal component analyses (PCAs) based on the relative abundances of the phyla (Appendix A) for both sampling campaigns are shown in Figure 4 and Figure 5, respectively. For both sampling dates, PC1 and PC2 explain about 54% of the variation. Although not a strong separation, the samples of the upper Ashalim strip (N), which was less affected by the spillage, tend to cluster with the control samples on the main axis (PC1).

### 3.3. Bacterial Diversity in the Biocrust Strip: Genus Level

The relative abundance of all detected genera (at least 1% in one of the strips of the streams) is given in Appendix A for the first and the second sampling campaigns, respectively. For the first sampling, the genus *Bryobacter* within the Acidobacteria was 2–4 times more abundant in the control stream strips than in the Ashalim strips. Within the Actinobacteria, the *Blastococcus* genus was more abundant in Ashalim strips compared to the control ones, and the abundance of *Rubrobacter* was comparable between the control and Ashalim samples, being higher in the upper strips in both streams. Within the Bacteriodetes, the genera *Flavisolibacter*, *Segetibacter* and *Rhodocytophaga* were more abundant in the control samples than in the corresponding samples from the Ashalim stream, while the genus *Pontibacter* showed the opposite trend. Within the *Firmicutes*, the abundance of the *Bacillus* genus was higher in the Ashalim samples than the corresponding control samples, particularly in the middle strips (D, L), reaching up to 15.3% for the L (bright soil) strip of the Ash2 site. Within the Proteobacteria, the abundance of the genus *Micovirga* was similar in the three control strips and significantly higher than the abundance in the samples of the lower strips (G, D) of the Ashalim stream. The genus *Rubellimicrobium* was more abundant in the control samples. Spatially, it was more abundant in stream bed samples than in the upper strips in both the control and Ashalim samples. The *Massilia* genus showed the highest difference between the control and Ashalim samples, reaching 10–16% relative abundance in the Ashalim stream bed compared to 0.52% in the control stream bed; the same trend was evident for the *Noviherbaspirillum* genus. Interestingly, although the *Nitrosomonas* genus (ammonia oxidizer) was detected in the control samples (0.1–0.4%), it was barely detectable in the Ashalim samples, the highest abundance being 0.05–0.12% in the stream bank samples (N strip).

The results of the second sampling showed that the genus *Bryobacter* remained more abundant in control (0.8–3.4%) than in Ashalim (0.02–0.65%) samples and the *Rhodocytophaga* genus showed high abundance in the lower strips of the control samples (2.5–3.9%) compared to all other samples. The *Blastococcus* and *Arthrobacter* genera were more abundant in the Ashalim samples than in the control samples. The *Pseudarthrobacter* and *Rubrobacter* genera were highly abundant in both the Ashalim and control streams, but without a clear trend between and within the control and Ashalim samples. The genera *LWQ8*, *TM7a* and *Saccharimonadales*, were more abundant in Ashalim samples than in the controls, and the *Bacillus* genus was barely detected in the control samples. The genera *AKIW781* and *Truepera* showed increasingly higher abundances toward the upper strips of the control and Ashalim streams. Within Proteobacteria, the *Sphingomonas* genus was more abundant in Ashalim samples compared to the controls, particularly in the lower strips, while the *Rubellimicrobium* genus was highly abundant, with the abundances being comparable between and within locations. Skermanella and *Devosia* were more abundant in Ashalim samples than the controls, with the abundances decreasing toward the upper strips. The *Microvirga* genus, although highly abundant, did not show clear patterns within and between samples. Interestingly, the *Nitrosospira* genus (ammonia oxidizer) was barely detected in the control but showed a measurable abundance in the Ashalim samples (0.04–1.43%), while the *Nitrobacter* genus (nitrite oxidizer) did not show a clear pattern; however, it also was generally higher in the Ashalim samples than in the controls.

### 3.4. Cyanobacteria and nifH Abundance

The diversity of the genera in the Cyanobacteria phylum was studied using 16S rRNA gene fragments amplified from the 2022 samples using cyanobacteria-specific PCR-primers. The filamentous *Leptolyngbya*, *Tychonema*, and *Trichocoleus* genera were the most dominant cyanobacteria in all samples (Table 2). The *Leptolyngbya* genus constituted 39–69% of the cyanobacteria in the control samples compared to 64–95% in Ashalim samples. In addition to *Scytonema*, the nitrogen-fixing *Nostoc* genus was detected in the streambed of the control stream and the three lower strips at site 2 of the Ashalim stream. The genera *Microcoleus* and *Nodosilnea* were detected mainly in the control stream. In contrast, the *Crinalium* genus was detected in all samples, and in some of the Ashalim strips its relative abundance was higher than that in the control strips. The genera *LWQ8*, *Aliterella*, *Loriellopsis* and *Vampirovibrio* were highly abundant in the strips of the two Ashalim sites compared to the corresponding strips of the control stream.

The abundance of the *nifH* gene was used as a marker for nitrogen fixation potential. The results in Table 3 show that in the control stream the highest abundance of *nifH* was measured in the streambed, which also matched the highest abundance of the *Cyanobacteria* phylum in this strip. This finding is possibly due to the accumulation of water in the lowest part of the stream during rare rain events, relative to the upper strips. The nitrogen-fixing genera *Scytonema* and *Nostoc* were also detected in this strip (Table 3). Within the Ashalim strips, the copy numbers of *nifH* were low and comparable and without any clear temporal or spatial trends. An exceptionally higher abundance of *nifH* was found in the stream bank strip of the Ash1 site during the second sampling.

## 4. Discussion

### 4.1. Bacterial Diversity: Phylum Level

Following exposure to acidic saline wastewater from the phosphate mining industry, the Ashalim biocrust samples exhibited higher electrical conductivity (EC) values compared with the control samples. The salinity of the samples declined by the time of the second sampling, likely due to the leaching of salts during rain events (Table 1). Similarly, during the first sampling, the pH of most Ashalim samples was lower than that of the control samples, whereas in the second sampling, pH values were comparable across all samples (Table 1). Both salinity and pH are key environmental parameters influencing soil microbial community structure in diverse ecosystems, including desert environments [40,41,42,43]. The high salinity and low pH values of Ashalim biocrust (Table 1) may explain their dominance by Actinobacteria during the first sampling (Appendix A). Actinobacteria are known to dominate biocrusts in dry hot deserts [43,44], and to survive extreme environmental conditions as spores [45]. This may also apply to the dominance of the mostly Gram-positive and spore forming Firmicutes in Ashalim samples. Bacteria of this phylum can survive dry hot conditions, such as the Atacama Desert [46,47]. The slow-growing Gram-negative Chloroflexi, usually found in freshwater and in nutrient-poor ecosystems [48,49,50], were more abundant in the lower strips of the control samples vs. the Ashalim samples; while in the higher strips (which were less affected by the contaminating event), the content of Chloroflexi was comparable in the control and Ashalim samples.

### 4.2. Bacterial Diversity: Genus Level

A summary of the dominant genera identified during the first and second samplings, highlighting differentially expressed genera, is presented in Table 4. In the control stream, 15 dominant genera (≥1% relative abundance in at least one sample) were shared between the two sampling campaigns, whereas 11 genera differed in their dominance. This variation likely reflects seasonal influences [51], as the first sampling occurred in winter and the second during the hot, dry summer. Similarly, in the Ashalim samples, 15 dominant genera were common to both samplings, while 22 genera exhibited differential dominance. In this case, beyond seasonal effects, the observed differences may also be associated with variations in pH and EC values (Table 1; [40,41,42,43]). Comparing between streams, 19 dominant genera were shared between the Ashalim and control samples, while nine genera showed differential abundance during the first sampling (Table 4). Of these, *Bryobacter and Segetibacter* were found in the control samples, while *Modestobacter*, *Kocuria*, *Marmoricola*, *Solirubrobacer*, *Bacillus*, *Paenibacillus*, *Pulluanibacillus*, were associated with Ashalim samples. During the second sampling, there were 19 genera common to the control and Ashalim samples, and eight genera exhibited differential abundance. *Segetibacter* was found only in control samples (both sampling dates), while *Kocuria*, *Nitrobacter*, *Bacillus*, *LWQ8*, *TM7a*, *Devosia*, and *Nitrosospira* were exclusive to the Ashalim samples. *Segetibacter* is a Gram-negative, strictly aerobic, heterotrophic, non-spore-forming genus [52] found in soil and fresh water, and thus it was not surprising that these bacteria were not detected in high abundance in the contaminated Ashalim samples. The *Bryobacter* genus, which was not dominant in Ashalim samples at the time of the first sampling, had become a dominant genus by the second sampling. This genus is known to inhabit acidic wetlands and soils [53], and therefore its high abundance in the low pH Ashalim samples was to be expected. Its absence in the first sampling at Ashalim may be attributed to the high salinity of contaminant. The *Bacillus* genus was highly abundant in the Ashalim samples (0–15.3%, differentially in strips), while in the control samples, its abundance was only 0.0–0.24% (Appendix A). This genus is made up of species that form endospores as a survival strategy in unfavorable environments, such as the contaminated Ashalim soil [46].

### 4.3. Cyanobacteria and the Potential for Nitrogen Fixation

Cyanobacteria are key components of biocrust formations, providing the ecosystem with fixed nitrogen and carbon. Filamentous cyanobacterial species are pioneer colonizers of biocrusts [54], forming the structural scaffold to which other organisms attach. They also contribute to soil aggregation through the secretion of extracellular polysaccharides that bind soil particles together [55]. The abundance of the Cyanobacteria phylum in control biocrust samples was up to tenfold higher than in the corresponding strips of the Ashalim samples (Table 2). The 2022 samples, however, indicated renewed cyanobacterial colonization, particularly at the Ash2 site. Notably, the Cyanobacteria phylum was most abundant in the stream bed across all sites, a strip that receives runoff water during sporadic rainfall events in this arid region.

Within the Cyanobacteria, the control biocrust samples were dominated by the filamentous non-diazotrophic genera *Leptolyngbya* (54–69%) and *Tychonema* (11–27%)*. Leptolyngbya* constituted the most dominant genus in all the Ashalim samples (84–95%) except for two strips where *Tychonema* exhibited an abundance of 24% for Ash1 and 15% for Ash2. A recently published review indicates that, globally, *Leptolyngbya* is the second most common biocrust taxon after the filamentous *Microcoleus vaginatus* [56]. Previous studies have indeed shown that *Microcoleus vaginatus* is the dominant filamentous genus at many sites in the Negev Desert [57,58]. However, a previous study of cyanobacterial diversity along a precipitation gradient in the northwest Negev Desert showed that *Microcoleus* and *Leptolyngbya* dominated the biocrust, and each constituted about 20% in the southern section of the Desert with an average annual rainfall of about 100 mm. However, the abundance of *Leptolyngbya* showed a substantial decline to 10 and 5.8% in the 130 mm and 170 mm sections of the gradient, respectively [59]. The average annual rainfall in the Ashalim region is about 70 mm, suggesting that the *Leptolyngbya* genus has adapted to the dry areas of the Negev Desert. In parallel, since it was highly abundant in Ashalim, almost as the sole Cyanobacteria genus, our findings imply that it is the pioneer cyanobacterial genus in this disturbed biocrust or possibly that it is more resistant to the contaminants than the other Cyanobacteria genera. The filamentous *Tychonema_CCAP_1459-11B* genus was the second most abundant Cyanobacteria genus, particularly in control samples (Table 2) and in some Ashalim strips. This genus has not previously been detected in the Negev Desert (using the same PCR primers as applied here) [15,59], indicating that the conditions of the study area are unique. The abundance of the *Trichocoleus* genus did not show a clear trend, and the highest abundance [3.5%] was detected in the dark soil strip at Ash1. The detection of this genus in most of the Ashalim and control samples, albeit with a low abundance, is in keeping with the findings of Hagemann et al. [59]. *Trichocoleus SAG26.92* was initially isolated from the Negev Desert, but the genus has also been detected in other hot deserts [56,60].

The diazotrophic cyanobacteria *Scytonema* and *Nostoc* [61] were detected only in a few samples and in small numbers (Table 2). This finding is reflected in the low, and comparable, abundance of *nifH* copies in all but three stream strips. Interestingly, the highest *nifH* numbers were measured in the stream bed strip of the control stream, coinciding with higher abundance of *Scytonema* and *Nostoc*. These two genera are known to dominate the nitrogen fixers in the Negev Desert [15,59,62]. The dominance of the filamentous cyanobacteria in the biocrust samples in both Ashalim and the control samples, in addition to the low abundance of nitrogen fixers, suggest that the biocrust in this hot dry area is in its early successional stages [16,31,63,64]. Biocrusts are the main source of fixed nitrogen in hot deserts [4]. indeed, the biocrust samples contained substantial amounts of inorganic nitrogen (ammonia and nitrate), with higher amounts in the Ashalim strips (Table 1). This nitrogen could have originated from the activity of nitrogen-fixing cyanobacteria, existing in low numbers in most samples. It is also possible that nitrogen could have been added to the biocrust through the deposition of atmospheric dust [65]. Finally, yet another source of inorganic nitrogen could be the activity of species of the free-living nitrogen-fixing *Azospirillum* genus [66], which were detected in all samples, albeit in small numbers (Appendix A).

The observed differences in the abundance and diversity of Cyanobacteria between the control and the contaminated Ashalim stream samples indicate the limited recovery potential of the Ashalim biocrust. The regeneration and establishment of disturbed biocrusts are known to be slow processes that may require 5–20 years [7]. These differences could result from the physical removal of the underdeveloped biocrust during the initial surge of contaminating water and the subsequent washing that followed the event. In addition, exposure to acidic wastewater [25] containing a complex mixture of heavy metals, such as lead, arsenic, nickel, molybdenum, copper, and chromium, likely contributed to delaying the successional colonization and development of the biological soil crust [67]. Notably, similarly low cyanobacterial abundances have been reported in biocrusts from restored post-mining sites in the Negev Desert, regardless of the duration since restoration began [68].

## 5. Conclusions

The results of this study demonstrate that even short-term exposure [as the area was washed within a few days by fresh water] to phosphate mining effluent can have long-lasting effects on the microbial community of desert biocrusts. The near disappearance of cyanobacteria from the biocrust layer five years after the contamination event suggests that ecosystem recovery is extremely slow and that full rehabilitation may require many years.

## Figures and Tables

**Figure 1 microorganisms-13-02580-f001:**
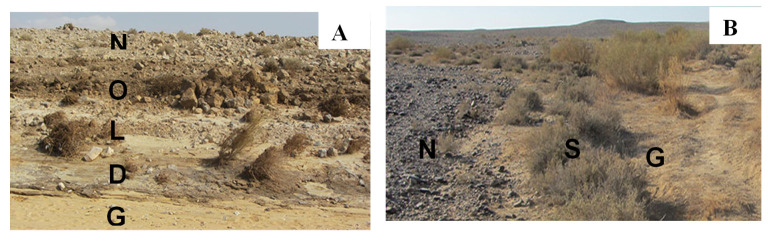
(**A**) Sampling site (Ash1) in the Ashalim stream: G, stream bed; D, dark soil surface; L, bright soil surface; O, organic matter surface; N, stream bank surface (mostly unflooded area). (**B**) Sampling site in the Gmalim stream: G, stream bed; S, shrub area; N, stream bank area (slope).

**Figure 2 microorganisms-13-02580-f002:**
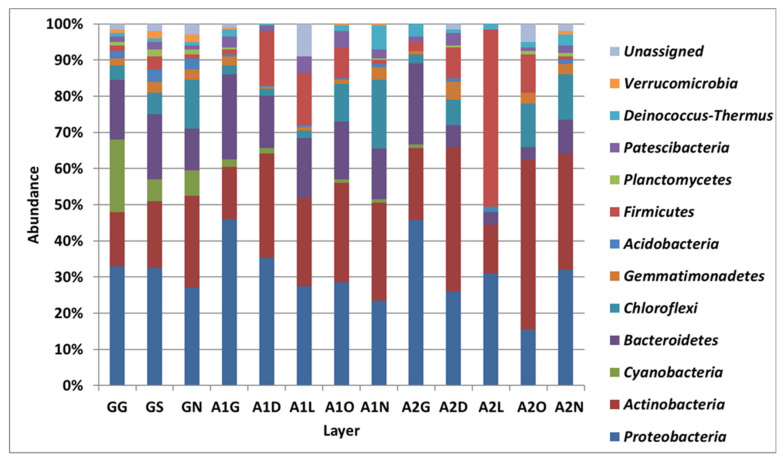
Relative abundance of each bacterial phylum detected during the 2018 sampling. Relative abundance was calculated by summing the clean sequences of all OTUs belonging to each phylum divided by the total number of all clean DNA sequences generated for the sample and multiplied by 100 (Appendix A). Values are averages of two NGS repeats for each strip, in which extracted DNA was combined from two biocrust samples. Sites: G, Gmalim; A1, Ashalim site 1; A2, Ashalim site 2. Ashalim Strips: G, stream bed; D, dark soil surface; L, bright soil surface; O, organic matter surface; N, stream bank surface. Gmalim strips: G, stream bed; S, shrub area; N, stream bank.

**Figure 3 microorganisms-13-02580-f003:**
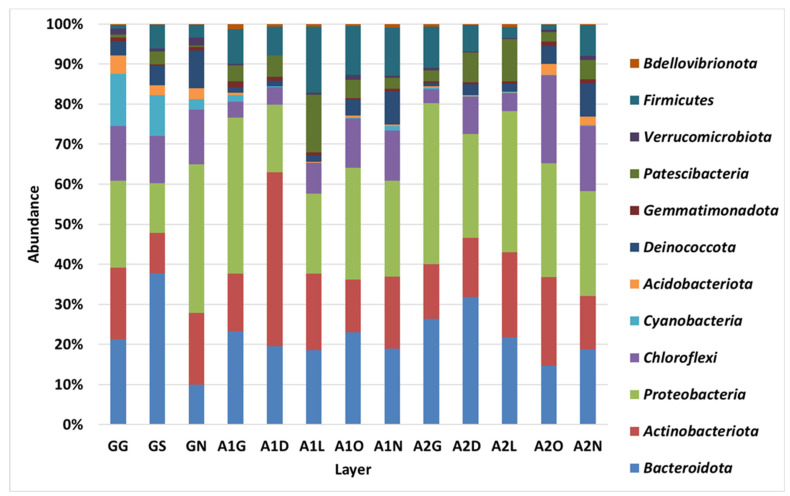
Relative abundance of each bacterial phylum detected during the 2022 sampling. Relative abundance was calculated by summing the clean sequences of all OTUs belonging to each phylum divided by the total number of all clean DNA sequences generated for the sample and multiplied by 100 (Appendix A). Values are averages of two NGS repeats for each strip, in which extracted DNA was combined from two biocrust samples. Sites: G, Gmalim; A1, Ashalim site 1; A2, Ashalim site 2. Ashalim Strips: G, stream bed; D, dark soil surface; L, bright soil surface; O, organic matter surface; N, stream bank surface. Gmalim strips: G, stream bed; S, shrub area; N, stream bank.

**Figure 4 microorganisms-13-02580-f004:**
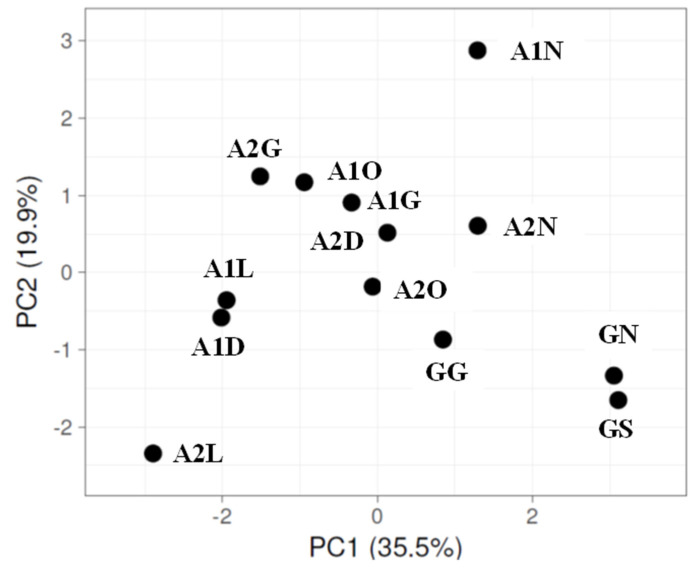
PCA for phylum relative abundance, generated using ClustVis2 [39], for 2018 samples. Based on the average of two NGS repeats for each strip (Appendix A). Extracted DNA was combined from two biocrust samples. Sites: G, Gmalim; A1, Ashalim site 1; A2, Ashalim site 2. Ashalim Strips: G, stream bed; D, dark soil surface; L, bright soil surface; O, organic matter surface; N, stream bank surface. Gmalim strips: G, stream bed; S, shrub area; N, stream bank.

**Figure 5 microorganisms-13-02580-f005:**
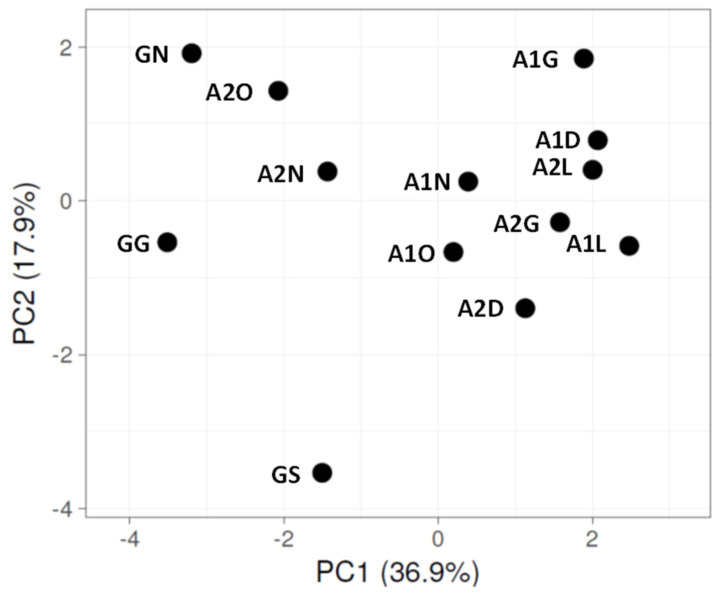
PCA for phylum relative abundance, generated using ClustVis2 [39], for 2022 samples. Based on the average of two NGS repeats for each strip (Appendix A). Extracted DNA was combined from two biocrust samples. Sites: G, Gmalim; A1, Ashalim site 1; A2, Ashalim site 2. Ashalim Strips: G, stream bed; D, dark soil surface; L, bright soil surface; O, organic matter surface; N, stream bank surface. Gmalim strips: G, stream bed; S, shrub area; N, stream bank.

**Table 1 microorganisms-13-02580-t001:** Chemical properties of the biocrust samples (top 1 cm soil layer). The samples were collected from two sites in Ashalim stream and one in Gmalim stream (control). Values are means ± SD of 4 samples (×2 duplicates) from each strip. Ashalim strips: G, stream bed; D, dark soil surface; L, bright soil surface; O, organic matter surface; N, stream bank surface (mostly unflooded area). Gmalim strips: G, stream bed; S, shrub area; N, stream bank area. Total ammonia nitrogen (TAN) and nitrate nitrogen in soil samples (mg-N/kg dry soil); EC, (μS^−1^).

	November 2018 Samplings				July 2022 Samplings			
	pH	EC	TAN	NO_3_^−^N	pH	EC	TAN	NO_3_^−^N
Site 1—Ashalim 1								
G	6.8 ± 0.0 ^b^ *	2014 ± 999 ^ab^	6.5 ± 1.5 ^b^	3.0 ± 2.0 ^b^	7.2 ± 0.1 ^b^	857 ± 111 ^a^	5.7 ± 8.1 ^ab^	14.2 ± 4.9 ^a^
D	7.0 ± 0.1 ^a^	1597 ± 814 ^b^	14.8 ± 9.4 ^ab^	66.8 ± 76.4 ^ab^	6.1 ± 0.2 ^c^	443 ± 299 ^bc^	14.2 ± 7.3 ^a^	4.0 ± 2.9 ^b^
L	7.0 ± 0.1 ^a^	856 ± 296 ^c^	16.1 ± 7.5 ^a^	23.3 ± 31.0 ^ab^	7.9 ± 0.2 ^ab^	239 ± 151 ^c^	13.7 ± 5.6 ^a^	3.0 ± 4.7 ^b^
O	6.7 ± 0.0 ^c^	3322 ± 481 ^a^	7.0 ± 4.6 ^ab^	12.0 ± 7.5 ^ab^	7.9 ± 0.1 ^ab^	479 ± 245 ^bc^	1.2 ± 1.8 ^b^	9.7 ± 7.0 ^ab^
N	7.1 ± 0.1 ^a^	1412 ± 539 ^bc^	6.6 ± 2.2 ^b^	55.5 ± 37.3 ^a^	8.3 ± 0.3 ^a^	415 ± 365 ^bc^	6.7 ± 5.7 ^b^	15.0 ± 20.2 ^ab^
Site 2—Ashalim 2								
G	7.1 ± 0.1 ^a^	1400 ± 606 ^b^	8.2 ± 2.1 ^ab^	2.5 ± 1.7 ^c^	7.8 ± 0.1 ^b^	531 ± 125 ^a^	2.7 ± 4.8 ^abc^	15.5 ± 5.4 ^a^
D	6.8 ± 0.0 ^b^	2457 ± 73 ^ab^	8.8 ± 2.6 ^ab^	28.7 ± 44.4 ^ab^	8.1 ± 0.3 ^ab^	319 ± 150 ^ab^	17.0 ± 11.5 ^a^	6.7 ± 6.4 ^ab^
L	6.6 ± 0.1 ^c^	2667 ± 176 ^ab^	12.6 ± 3.0 ^a^	9.9 ± 8.8 ^abc^	8.0 ± 0.5 ^ab^	395 ± 386 ^ab^	8.5 ± 5.1 ^ab^	7.0 ± 2.3 ^b^
O	6.9 ± 0.2 ^b^	3237 ± 849 ^a^	2.9 ± 2.5 ^c^	22.8 ± 9.3 ^ab^	8.4 ± 0.0 ^a^	142 ± 28 ^b^	6.5 ± 3.6 ^ab^	16.2 ± 9.6 ^ab^
N	7.1 ± 0.1 ^a^	668 ± 166 ^c^	5.0 ± 2.2 ^bc^	20.6 ± 12.1 ^ab^	8.2 ± 0.2 ^ab^	334 ± 151 ^ab^	0.25 ± 0.5 ^c^	33.0 ± 32.3 ^abc^
Control—Gmalim								
G	6.4± 0.0 ^c^	112 ± 28 ^c^	0.1 ± 0.0 ^a^	2.4 ± 1.4 ^b^	8.3 ± 0.2 ^b^	262 ± 210 ^a^	1.75 ± 0.95 ^b^	4.3 ± 1.5 ^b^
S	8.75 ± 0.2 ^a^	468 ± 276 ^b^	1.5 ± 1.9 ^a^	33.8 ± 23.8 ^a^	8.9 ± 0.2 ^a^	209 ± 79 ^a^	1.5 ± 2.6 ^b^	31.7 ± 12.2 ^a^
N	8.15 ± 0.3 ^b^	1581 ± 1080 ^a^	0.6 ± 1.2 ^a^	27.2 ± 11.3 ^a^	8.6 ± 0.4 ^ab^	267 ± 191 ^a^	8.75 ± 3.8 ^a^	13.0 ± 10.4 ^ab^

* Columns with different lowercase letters are significantly different (*p* < 0:05). One-way ANOVA and Tukey Honest Significant Difference (HSD) tests, and a normality test (JMP statistical software, V14, were used to test significant differences between the mean values of the analysis.

**Table 2 microorganisms-13-02580-t002:** Abundance (% of total sequences in each sample) of assigned cyanobacterial genera using *16S rRNA* gene cyanobacteria-specific PCR primers. Percentages were calculated by dividing the clean sequences of each identified cyanobacterial genus by the total sequences of all assigned cyanobacterial genera and multiplying by 100. Cyanobacterial 16S rRNA gene fragments were amplified from a total DNA mixture extracted from 4 separate biocrust samples taken from each strip. OM—organic matter. Abundance (%) of *Cyanobacteria* phylum was taken from Appendix A.

	Control—Gmalim			Site 1-Ashalim 1					Site 2-Ashalim 2				
Genus	Stream Bed	Shrubs	Stream Banks	Stream Bed	Dark Soil	Bright Soil	OM Foam	Stream Banks	Stream Bed	Dark Soil	Bright Soil	OM Foam	Stream Banks
**Relative abundance of *Cyanobacteria* phylum**													
**2018**	20.0	6.0	7.0	2.0	1.5	0	1.0	1.0	1.0	0	0	0	0
**2022**	13.0	10.1	2.5	1.6	0.2	0.1	0.2	1.1	0.4	0.17	0.2	0.1	0.3
**Relative abundance of *Cyanobacteria* genera (2022)**													
** *Leptolyngbya* **	54.64	39.97	69.37	93.62	67.9	88.5	94.23	94.21	91.85	95.29	84.37	64.32	91.47
** *Tychonema_CCAP_1459-11B* **	11.27	18.09	27.02	2.44	23.93	0.44	0.76	0.70	0.92	2.11	0.81	15.54	0.07
** *Trichocoleus_SAG_26.92* **	1.25	1.83	0.54	0.64	3.59	0.00	0.63	2.13	0.28	0.09	1.68	2.69	3.10
** *Nostoc* **	0.21	0.00	0.00	0.00	0.00	0.00	0.00	0.00	0.12	0.09	0.10	0.00	0.00
** *Scytonema* **	1.03	0.00	0.00	0.00	0.00	0.54	0.00	0.00	0.00	0.00	0.00	0.00	0.00
** *Symplocastrum_* **	1.85	0.06	0.43	0.01	0.00	0.00	0.00	0.00	0.00	0.09	0.00	0.00	0.00
** *CENA518* **	8.87	0.01	0.00	0.00	0.00	0.00	0.00	0.00	0.00	0.00	0.00	0.00	0.00
** *Calothrix_PCC-6303* **	0.00	0.00	0.00	0.00	0.00	0.00	0.00	0.00	0.00	0.00	0.00	0.49	0.00
** *Sericytochromatia* **	0.00	0.01	0.01	0.00	0.00	0.00	0.03	0.01	0.08	0.03	0.00	0.61	0.05
** *Microcoleus* **	0.16	0.06	0.03	0.00	0.00	0.00	0.00	0.00	0.00	0.00	0.00	0.00	0.00
** *Nodosilinea_* **	14.11	0.07	0.00	0.00	0.00	0.00	0.00	0.00	0.00	0.00	0.00	0.00	0.15
** *Crinalium* **	2.64	0.33	1.16	0.29	0.00	5.48	0.53	0.61	1.15	0.00	1.95	8.32	0.39
** *LWQ8* **	0.00	0.04	0.00	0.00	4.08	0.98	0.12	0.04	0.00	0.04	1.68	2.02	0.39
** *Aliterella* **	0.15	0.00	0.00	0.01	0.00	0.28	0.43	0.16	0.28	0.07	0.75	1.16	0.22
** *Pleurocapsa* **	0.00	0.03	0.07	0.00	0.00	0.00	0.28	0.25	0.00	0.00	0.00	0.12	1.14
** *Loriellopsis* **	0.22	0.07	0.27	0.10	0.00	0.35	0.61	0.48	0.00	1.00	0.00	1.53	0.19
** *Vampirovibrio* **	0.00	0.00	0.00	0.01	0.67	0.09	0.12	0.04	0.23	0.03	0.06	0.12	0.10
**Uncultured**	3.37	39.39	1.02	2.66	0.00	3.33	2.14	1.29	4.87	1.15	8.03	1.52	2.28

**Table 3 microorganisms-13-02580-t003:** Enumeration of *nifH* gene (copies g^−1^ dry soil); values are means ± SD of 5 × 2 repeats. Ashalim strips: G, stream bed; D, dark soil surface; L, bright soil surface; O, organic matter surface; N, stream bank surface (mostly unflooded area). Gmalim strips: G, stream bed; S, shrub area; N, stream bank area. Abundance (%) of *Cyanobacteria* phylum was taken from Appendix A.

Layer	*nifH* 2018	*Cyanobacteria* 2018	*nifH* 2022	*Cyanobacteria* 2022
**Ashalim 1**				
**G**	31,300 ± 3535	2.0	5500 ± 2380	1.6
**D**	27,700 ± 39,173	1.5	5750 ± 2500	0.2
**L**	30,429 ± 43,033	0	7250 ± 3095	0.1
**O**	-	1.0	4252 ± 3496	0.2
**N**	22,228 ± 31,436	1.0	106,000 ± 19,602	1.1
**Ashalim 2**				
**G**	16,630 ± 23,518	1.0	13,250 ± 17,876	0.4
**D**	41,675 ± 10,720	0	5000 ± 1825	0.17
**L**	45,712 ± 12,444	0	3100 ± 3450	0.2
**O**	17,490 ± 24,734	0	1375 ± 750	0.1
**N**	29,296 ± 3247	0	14,000 ± 10,708	0.3
**Gmalim**				
**G**	157,462 ± 15,084	20.0	518,000 ± 98,813	13.0
**S**	28,100 ± 19,657	6.0	1575 ± 1650	10.1
**N**	7900 ± 11,172	7.0	6500 ± 3696	2.5

**Table 4 microorganisms-13-02580-t004:** Dominant (at least 1% relative abundance in one of the strips) and differentially abundant genera in Ashalim and control streams at the two sampling events (abundance is given in Appendix A). Differential genera are written in bold and underlined. *Cyanobacteria* genera are not included.

	November 2018	July 2022
**Dominant genera in the Ashalim stream**	*Blastococcus*, *Geodermatophilus*, ***Modestobacter***, *Arthrobacter*, ***Kocuria***, *Pseudoarthrobacter*, ***Marmoricola***, *Rubrobacter*, ***Solirubrobacer***, *Flaviaesturariibacter*, *Rhodocytophaga*, *Adhaeribacter*, *Hymenobacter*, *Pontibactor*, *Truepera*, ***Bacillus***, ***Paenibacillus***, *Planomicrobiom*, ***Pulluanibacillus***, *Skermanella*, *Microvirga*, *Rubellimicrobium*, *Ellin6055*, *Sphingomonas*, *Massilia*, *Noviherbaspirillum.*	*Bryobacter*, *Blastococcus*, *Arthrobacter*, ***Kocuria***, *Pseudoarthrobacter*, *Rubrobacter*, *Flaviaesturariibacter*, *Rhodocytophaga*, *Adhaeribacter*, ***Nitrobacter***, *Pontibactor*, *AKIW781*, *AKYG1722*, *JG30-KF-CM45*, *Truepera*, ***Bacillus***, *Planococcus*, ***LWQ8***, ***TM7a***, *Saccharimonadales*, *Skermanella*, *Microvirga*, ***Devosia***, *Rubellimicrobium*, *Sphingomonas*, *Nitrosospira.*
**Dominant genera in the control stream**	***Bryobacter***, *Blastococcus*, *Geodermatophilus*, *Arthrobacter*, *Pseudoarthrobacter*, *Rubrobacter*, *Flaviaesturariibacter*, ***Segetibacter***, *Rhodocytophaga*, *Adhaeribacter*, *Hymenobacter*, *Pontibactor*, *Truepera*, *Planomicrobiom*, *Skermanella*, *Microvirga*, *Rubellimicrobium*, *Ellin6055*, *Sphingomonas*, *Massilia*, *Noviherbaspirillum*	*Bryobacter*, *Blastococcus*, *Arthrobacter*, *Pseudoarthrobacter*, *Rubrobacter*, *Flaviaesturariibacter*, ***Segetibacter***, *Rhodocytophaga*, *Adhaeribacter*, *Pontibactor*, *AKIW781*, *AKYG1722*, *JG30-KF-CM45*, *Truepera*, *Planococcus*, *Saccharimonadales*, *Skermanella*, *Microvirga*, *Rubellimicrobium*, *Sphingomonas*

## Data Availability

The original contributions presented in this study are included in the article/Appendix A. Further inquiries can be directed to the corresponding author.

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
