# Peer review of "Effect of Contamination by Phosphate Mining Effluent on Biocrust Microbial Community Structure and Cyanobacterial Diversity in a Hot Dry Desert"

_microorganisms, 2025, doi:10.3390/microorganisms13112580_

Round 1
Reviewer 1 Report
Comments and Suggestions for Authors
- While the current findings are interesting, the relatively small sample size of data points may limit the statistical power. We recommend expanding the dataset through additional replicates, longer time-series, or more sampling sites to reinforce the conclusions. What about the fungal community?
- The literature reviewed in the Introduction appears dated. The authors should incorporate more recent references to reflect current advancements in the field.
- The Materials and Methods section is overly simplistic. Please provide a more detailed description, including specifics on data processing and analysis.
- Table 1 lacks statistical analysis of differences between groups. Please add appropriate significance testing (e.g., ANOVA or t-tests with p-values) to support the reported variations.
- The resolution of Figure 2 is too low for proper interpretation. We recommend separating it into two distinct figures and increasing the font size for better readability.
- Figure 3 has the same issues as previously mentioned for Fig. 2.
Author Response
Comments and Suggestions for Authors
- While the current findings are interesting, the relatively small sample size of data points may limit the statistical power. We recommend expanding the dataset through additional replicates, longer time-series, or more sampling sites to reinforce the conclusions. What about the fungal community?
Response: After the contamination event, several specialist research groups were invited to study the affected ecosystem, addressing the hydrological and physical consequences of the incident, as well as various biological components (microorganisms, plants, wild animals, birds, insects, etc.). Our research focused on the bacterial diversity of biocrusts, while other aspects such as fungi and nematodes were investigated by other groups. However, based on our findings—particularly the delayed recovery of cyanobacteria—and the complementary results from the other teams, the authorities decided to replace the soil in the sandy section of the contaminated stream. Consequently, we were unable to continue with further sampling campaigns.
- The literature reviewed in the Introduction appears dated. The authors should incorporate more recent references to reflect current advancements in the field.
Response: We selected the cited literature based on its relevance to our case study rather than its publication date. Nevertheless, our reference list includes over 60 works, most of which were published within the last decade. If the reviewer insists on the inclusion of a specific reference, we will be more than willing to incorporate it.
- The Materials and Methods section is overly simplistic. Please provide a more detailed description, including specifics on data processing and analysis.
Response: We carefully reviewed the Materials and Methods section and found that it already includes detailed information such as volumes, concentrations, durations, and the original references. However, if the reviewer specifies a particular section that requires revision, we will be more than willing to make the necessary changes.
- Table 1 lacks statistical analysis of differences between groups. Please add appropriate significance testing (e.g., ANOVA or t-tests with p-values) to support the reported variations.
Response: statistical analysis was added to the table.
- The resolution of Figure 2 is too low for proper interpretation. We recommend separating it into two distinct figures and increasing the font size for better readability.
Response: Done. The figures were separated with bigger font size. Hope it is clearer now.
- Figure 3 has the same issues as previously mentioned for Fig. 2.
Response: The figures were separated with bigger font size. Hope it is clearer now.

Reviewer 2 Report
Comments and Suggestions for Authors
The impact of industrial and mining operations are particularly likely to affect environmental health and stability. This manuscript provides important evidence of how such human impacts may affect particularly fragile ecosystems such as the desert microbiome and biocrust communities. This manuscript focuses on desert ecosystems where there may have been less focus previously. There is a combination of environmental and biotic sources of evidence that makes it particular informative. Overall, this is a well organized and clearly written manuscript. I found no major issues. However, I noticed the following small corrections.
Line 308. The cited tables should be Tables S2 and S3, not S1 and S2.
Line 321. Again the cited tables should be Tables S2 and S3. I checked the data reported for cyanobacterial percentages and the text agrees with the proper Tables S2 and S3.
Please check your labeling of Figure 1B. The rocky, grey gully appears to be the stream bed, that you currently label as "N" stream bank, it should be labeled as G (?), and the downward sloping soil to the right labeled "G" appears to be the stream bank and should be labeled as N (?).
The evidence gathered clearly addresses how a control environment and the impacted environment provide insight into the effect of mining on the fragile biocrust communities.
The references are appropriate.
Author Response
Comments and Suggestions for Authors
The impact of industrial and mining operations are particularly likely to affect environmental health and stability. This manuscript provides important evidence of how such human impacts may affect particularly fragile ecosystems such as the desert microbiome and biocrust communities. This manuscript focuses on desert ecosystems where there may have been less focus previously. There is a combination of environmental and biotic sources of evidence that makes it particular informative. Overall, this is a well organized and clearly written manuscript. I found no major issues. However, I noticed the following small corrections.
Line 308. The cited tables should be Tables S2 and S3, not S1 and S2.
Response: Thanks- we checked this through the manuscript and corrected accordingly.
Line 321. Again the cited tables should be Tables S2 and S3. I checked the data reported for cyanobacterial percentages and the text agrees with the proper Tables S2 and S3.
Response: Thanks- we checked this through the manuscript and corrected accordingly.
Please check your labeling of Figure 1B. The rocky, grey gully appears to be the stream bed, that you currently label as "N" stream bank, it should be labeled as G (?), and the downward sloping soil to the right labeled "G" appears to be the stream bank and should be labeled as N (?).
Response: Thanks for the comment. The G is exactly the water path during rainfall event. Besides is where shrub grow and N is in the slope.
The evidence gathered clearly addresses how a control environment and the impacted environment provide insight into the effect of mining on the fragile biocrust communities.
The references are appropriate.

Round 2
Reviewer 1 Report
Comments and Suggestions for Authors
I think the current format is ready for publication.